# Human Neural Stem Cell-Based Drug Product: Clinical and Nonclinical Characterization

**DOI:** 10.3390/ijms232113425

**Published:** 2022-11-03

**Authors:** Daniela Celeste Profico, Maurizio Gelati, Daniela Ferrari, Giada Sgaravizzi, Claudia Ricciolini, Massimo Projetti Pensi, Gianmarco Muzi, Laura Cajola, Massimiliano Copetti, Emilio Ciusani, Raffaele Pugliese, Fabrizio Gelain, Angelo Luigi Vescovi

**Affiliations:** 1Unità Produttiva per Terapie Avanzate, Fondazione IRCCS Casa Sollievo della Sofferenza, 71013 San Giovanni Rotondo, Italy; 2Laboratorio Cellule Staminali, Cell Factory e Biobanca, AOSP S. Maria, 05100 Terni, Italy; 3Dipartimento di Biotecnologie e Bioscienze, Università Milano Bicocca, 20126 Milano, Italy; 4Unità di Biostatistica, Fondazione IRCCS Casa Sollievo della Sofferenza, 71013 San Giovanni Rotondo, Italy; 5Laboratorio Analisi, Instituto Nazionale Neurologico C. Besta, 20133 Milano, Italy; 6Unità di Ingegneria Tissutale, Fondazione IRCCS Casa Sollievo della Sofferenza, 71013 San Giovanni Rotondo, Italy

**Keywords:** neural stem cells, GMP, standardization, ATMP production, quality control

## Abstract

Translation of cell therapies into clinical practice requires the adoption of robust production protocols in order to optimize and standardize the manufacture and cryopreservation of cells, in compliance with good manufacturing practice regulations. Between 2012 and 2020, we conducted two phase I clinical trials (EudraCT 2009-014484-39, EudraCT 2015-004855-37) on amyotrophic lateral sclerosis secondary progressive multiple sclerosis patients, respectively, treating them with human neural stem cells. Our production process of a hNSC-based medicinal product is the first to use brain tissue samples extracted from fetuses that died in spontaneous abortion or miscarriage. It consists of selection, isolation and expansion of hNSCs and ends with the final pharmaceutical formulation tailored to a specific patient, in compliance with the approved clinical protocol. The cells used in these clinical trials were analyzed in order to confirm their microbiological safety; each batch was also tested to assess identity, potency and safety through morphological and functional assays. Preclinical, clinical and in vitro nonclinical data have proved that our cells are safe and stable, and that the production process can provide a high level of reproducibility of the cultures. Here, we describe the quality control strategy for the characterization of the hNSCs used in the above-mentioned clinical trials.

## 1. Introduction

Neural stem cells (NSCs) were discovered in 1992 by Reynolds and Weiss [1], who isolated them from the striatum of the adult mouse brain and reported the first use of epidermal growth factor (EGF) to induce NSC proliferation in vitro; later on, other groups showed that EGF and basic fibroblast growth (bFGF) factor cooperatively induced proliferation, self-renewal and expansion of NSCs obtained from mammal tissue; these NSCs are able to form spherical clusters called neurospheres [2,3]. Since those first discoveries, researchers have been working on the development of a hNSC-based therapy to treat neurodegenerative diseases. Preclinical data have shown that NSCs are able to counteract the pathological processes of many neurodegenerative pathologies. NSCs differentiate into mature cells that integrate into the nervous tissue, replacing damaged elements and restoring the functionality of the neural circuitry [4]. Open-label clinical trials investigating transplantation of ventral mesencephalic tissue from aborted human fetuses into striatum have provided proof of concept that cell replacement can reverse motor symptoms in some Parkinson’s disease (PD) patients [5,6,7,8]. In addition, more recent studies have shown the safety and the feasibility in the clinical use of neural stem cells in different neurodegenerative diseases such as amyotrophic lateral sclerosis (ALS) [9,10,11] or Huntington’s disease [12]. Recent data have shown that the effects produced by NSC transplants are not limited to the de novo generation of neurons and glial cells. A key element of their therapeutic ability consists of the secretion of neurotrophic and immunomodulatory factors, including those trafficked within extracellular membrane vesicles (EVs) [13], which preserve damaged cells and counteract intrinsic pathological events of the disease; this mechanism has been called “the paracrine hypothesis” or “the bystander effect” [4,14]. Translation of cell therapies, particularly those based on hNSCs, into clinical practice requires the adoption of robust production protocols in order to optimize and standardize the manufacture and cryopreservation of the cells, in compliance with good manufacturing practice (GMP) regulations. The first part of the process includes the standardization of raw material procurement and its characteristics, and the drafting of standard operating procedures for the isolation and expansion of target cells from the tissue source. The origin of the cells is the first issue that must be addressed to enable the application of NSCs in the clinical treatment. The reasons are that: (i) NSCs derived from fetal tissues raise ethical concerns, and (ii) the cell dose required for adequate patient treatment is usually very high. For example, the pioneer studies performed in PD patients [5] entailed the use of multiple fetal donors to treat a single patient in order to obtain the correct number of dopaminergic neurons for the transplant. The process to obtain and expand hNSCs requires extensive cell manipulation; according to the European Regulation (ER) 1394/2007, due to the long-time culture of our stem cells, they are legally classified as advanced therapy medicinal products (ATMPs), and as such, they are required to be produced in full compliance with good manufacturing practice standards. Following the coming-into-force of the ER 1394/2007, several guidelines have been published by the European Medicine Agency and the International Conference on Harmonization, because although the European regulatory framework establishes what kind of product characteristics must be investigated during nonclinical and clinical development, it does not clearly define how each characteristic should be analyzed. Because of the great variety of potentially different products, only general guidelines have been made available. Therefore, it is the responsibility of each investigator to define the characterization strategy for the investigated product and to make sure that such strategy is in compliance with the aforementioned guidelines. An extensive characterization of the cellular component should be established in terms of identity, purity, potency, viability, safety and suitability for the intended use as defined in the Guideline on human cell-based medicinal products (EMEA/CHMP/410869/2006). This guideline specifically acknowledges that conventional nonclinical pharmacology and toxicology studies may not be appropriate for these cell-based products; the intended quality of the drug substance should be determined through consideration of its use in the drug product as well as from the knowledge and understanding of its physical, chemical, biological and microbiological properties or characteristics, which can influence the development of the drug product (EMA/CHMP/ICH/425213/2011). Here, we show a thorough characterization of human neural stem cell lines obtained from human fetuses, expanded using the neurosphere technique [15,16] and according to the GMP guidelines, demonstrating that the hNSC lines maintain stable growth potential, differentiation ability and karyotype stability even after extensive manipulation and culture.

The procedures described in the following paragraphs have proved to be highly reliable and allowed us to conduct two hNSCs-based phase I clinical trials on 18 amyotrophic lateral sclerosis patients (EudraCT 2009-014484-39, NCT01640067) and 15 secondary progressive multiple sclerosis patients (EudraCT 2015-004855-37, NCT03282760). Since data from the latter clinical trial are currently being evaluated, here we will primarily discuss the details concerning the cell-based drug used in the ALS trial.

## 2. Results

### 2.1. Identity

In the first five years of activity (2010–2015), our facility received forty-three tissue samples. Fifteen of them were found noncompliant with starting material specification and rejected. We started twenty-eight productions, of which eighteen were suspended because of lack of cell growth, tissue contamination or genomic abnormalities. The remaining tissues originated ten released batches of intermediate product. Samples collected from fetuses younger than 15 gestational weeks resulted in smaller batches (fewer than thirty frozen vials for each one), while from older fetuses, we were able to obtain batches of up to one hundred and fifty vials for each one. 

For the above-mentioned ALS trial (EudraCT 2009-014484-39), a total number of 18 different drug product (DP) batches were produced, starting from 2 intermediate product (IP) batches obtained from two different tissue samples collected from fetuses that died between 15 and 16 gestational weeks. Histological examination confirmed the subventricular origin of the specimen retrieved for cell production (Figure 1a); moreover, as shown in Figure 1b, the investigation of the positional identity confirmed that the samples belonged to the same brain region (Appendix A Appendix A). 

All the cell batches, as expected, were able to differentiate in astrocytes, neurons and oligodendrocytes. Data collected from all the productions performed in our facility show that the population obtained with our differentiation protocol is composed by 44.57 ± 11.99% astrocytes, 24.91 ± 11.14% neurons and 20.75 ± 7.98% oligodendrocytes (See Appendix A). 

### 2.2. Safety 

#### European Pharmacopeia, Genomic Stability and Adventitious Virus Tests

All the drug products used in this study were found to be sterile and mycoplasma- and HSV1/2-free, with an endotoxin content less than 1 EU/mL. Karyotype analysis had been performed at four different critical control points for each donor: primary culture, intermediate product release, second-to-last passage before DP formulation, and DP release. Results were evaluated after counting at least 60 metaphases for each test. Data showed no structural or numeric abnormalities, and SNP array did not show any balanced microrearrangements.

### 2.3. Biological Tests

#### 2.3.1. In Vitro

Following the drug product formulation, a sample of each cell line was plated without the growth factor, as previously described [10]. All the lines showed incapacity of self-replication in the absence of EGF and bFGF, and cultures were extinct in 2–6 passages (18–53 days).

#### 2.3.2. In Vivo

All animal care and experimental procedures were carried out according to the current national and international animal ethics guidelines, and were approved by the Italian Ministry of Health (authorization number 651/2016-PR).

The multipotency, nontumorigenicity and biodistribution of the hNSCs were evaluated by long-term transplantation studies in athymic nude mice, which are ideal hosts for established, rapidly growing tumor cell lines. The hNSCs lines were unilaterally transplanted into the dorsal striatum (300.000 cells/mice). Histological analyses were performed at 6 months post-transplantation.

During the 6 months of clinical evaluation, we never observed clinical symptoms of suffering or hydrocephaly, weight loss or motor deficits such as ataxia due to rejection symptoms or mass overgrowth into the brain. Although all cell lines successfully integrated in all brain regions, they preferentially reached the striatum and the corpus callosum; however, each line displayed a peculiar pattern of distribution. The tested hNSC lines were prone to migrate away from the graft core reaching additional brain regions (Figure 2), including SVZ, showing a striking tropism for the niche, septum and cortex or even migrating contra laterally through the corpus callosum.

In order to determine the fraction of proliferating cells in situ, we performed immunostaining for the human proliferation marker Ki67. The hNSC lines never showed signs of tumorigenic transformation in long-term experiments and the proliferating cell fraction never exceeded 7% (Appendix A).

In these animals, on the one hand, the hNSCs showed a consistent migration ability throughout the brain (up to 6 mm at 6 months after transplant); on the other hand, we did not detect cells into randomly analyzed organs (heart, spleen, liver, lungs, kidney and lymph nodes up to 6 months) (see Figure 3). 

### 2.4. Purity 

To examine the lack of culture medium residues, which are considered impurities in the HBSS cell suspension (final product—FP), we set the retention times (RT) of complete medium (positive control), HBSS (negative control), culture medium exposed to the cells (worn out), first wash, and second wash. Impurities were determined via HPLC on an analytical C18 column using AcN and H_2_O, with 0.1% TFA (see Methods for further details). The chromatographic peaks of all samples were obtained with flow rates of 1 mL min^−1^ analyzing both λ = 240 nm and λ = 257 nm, as depicted in Figure 4. As expected, the different RT detected in the medium sample indicated a heterogeneous composition of hydrophilic and hydrophobic compounds. Indeed, three major fractions were observed with an RT of 6.47 min, 10.36 min, and 28.28 min (Figure 4A). On the contrary, HBSS gave a characteristic baseline over the 50 min of analysis, typical of an aqueous solution (Figure 4B). In worn-out medium, RT values were similar to those observed in the medium sample, except for the RT at 28.28 min, suggesting decreasing amounts of hydrophilic fraction (Figure 4C). In stark contrast, the first wash sample showed dramatic differences in the chromatographic peaks (Figure 4D). Although it had a similar HPLC RT to the worn-out medium (8.56 and 12.21 min, respectively), it revealed a sharp decrease in the intensity of the peaks (∼87.5% less). This large difference in spectra must therefore be attributed to the washings performed on the cells, which are important either to their purity and their biological activities. Lastly, the second wash sample showed a drastic change in the shape of the HPLC spectrum (Figure 4E) compared to both worn-out medium and first wash samples, featuring a trend similar to the one observed in the HBSS. These dramatic changes suggest that this reliable procedure allows us to obtain a drug with ≥98% purity without the presence of undesired fractions derived from the culture medium.

### 2.5. Potency

#### 2.5.1. In Vitro 

Figure 4 shows the growth rate of all the cell drug products administered to patients. Since there is no universal standard for the evaluation of the growth rate for hNSCs, for the purposes of GMP production we developed our own acceptability range. After an extensive data collection in our research laboratories, we identified two different clusters of lines: one with lines having a constant and progressive growth, the other one with lines showing no or really slow replication rates (often associated with karyotype alteration). Data obtained from the culture of the hNSCs were compared with the growth curve of cancer stem cells isolated from cerebral tumors (glioblastoma), kindly provided by our colleague Elena Binda. In order to avoid the selection of potential genetically modified cell lines, statistical analysis identified the optimum slope range as the one comprised between 0.025 and 0.165. All the DP batches produced for the clinical trial showed a growth curve slope comprised in the selected range. 

In vitro potency was confirmed using the clonal efficiency test which showed that even in the “worst case” scenario, when the cells were really diluted, our product retained its clonogenic potential. The principle of our technique is seeding cells in the complete medium at a concentration ten times lower than the normal culture (10^3^ cells/cm^2^ vs. 10^4^ cells/cm^2^), thus lessening the effects of cell-to-cell communication and the influence of cell-produced molecules in order to verify the single cell’s ability to proliferate. For this test, our in-house release specifications require that at least 1% of the seeded cells give rise to a cell aggregate (secondary clone) called the neurosphere. All the cell lines analyzed complied with the specifics. 

#### 2.5.2. In Vivo

An in vivo potency test was described in our previous work, which shows compelling evidence that intraspinal delivery of hNSCs ameliorates the course of the disease, delays the deterioration of motor functions and extends overall survival in a transgenic rat model of ALS (See [17], for complete results). 

## 3. Discussion

It took more than twenty years to move forward from the discovery of hNSCs and the theoretical concept of their use for the treatment of neurodegenerative diseases to their actual translation into the clinical application. 

This delay may be ascribed to scientific and technical challenges but also to regulatory ones. In the first decade of the 2000s, a profound revolution happened in the field of regenerative medicine, and the regulatory landscape became progressively more restricted in order to ensure patients’ safety. Specifically, in 2008 the ER 1394/2007 entered into force. Consequently, advanced-therapy medicinal products were described as new medical products based on genes (gene therapy), cells (cell therapy) and tissues (tissue engineering). These products require a long preparation time as well as extensive manipulation before they can be administered to patients. Because of the ER 1394/2007, since 2008 it has been mandatory in Europe that ATMPs are produced in cell factories, following the GMP rules.

Since our group started its research on hNSCs in the early ‘90s, we have had at our disposal a huge amount of scientific data that have needed to be organized and classified in order to fulfill the regulatory requirements to move on to clinical trials. Furthermore, it was necessary to define specific acceptance criteria for each test. More specifically, what was previously defined as compliant or not-compliant according to the researcher’s experience had to be transformed into a specific quantifiable parameter. Beyond the tests performed in accordance with the pharmacopoeia, we standardized methods for the definition of parameters for the stable expansion of hNSC lines. Our system provides a standard amplification process that enables us to produce identical hNSC lots from a single tissue specimen, in large enough quantities so that they can be transplanted into a large number of patients, allowing us to treat these patients with exactly the same, extremely standardized, cellular drug. This, along with stable dependence on growth factors, karyotype stability and lack of tumorigenicity, yields a functional and safety profile that warrants the GMP certification of hNSCs.

Our quality control strategy has been evaluated multiple times by the competent regulatory agency and has proved to be reliable and consistent through the clinical trials we have conducted in the last years. Moreover, the analyses carried out during the production process and during the quality control cycle performed during the ALS clinical trial confirmed that the hNSCs prepared using our GMP protocol are safe for administration to patients; indeed, results of compendial tests showed that all 18 batches conformed to specifics in terms of sterility and absence of Mycoplasma species, endotoxins and adventitious viruses. To the best of our knowledge, this is the first time that a GMP procedure for the isolation of hNSCs has been successfully carried out from fetuses that were certifiably deceased in utero due to natural causes; the choice of spontaneous abortion as a source of stem cells, in addition to eliminating ethical problems, provides some practical advantages based on the standardization of sampling procedures and an easier screening of donors. On the other hand, the use of tissue collected from miscarriages has some difficulties related to the scarcity of specimens in compliance with the quality criteria dictated by the law and the unpredictability of the spontaneous abortion that requires a quick and agile response of the cell factory team. Moreover, especially in samples obtained from fetuses younger than 15 gestational weeks, the amount of starting material can affect the performance of the process. Higher numbers of cells produced from the same donor are required for trials with large cohorts of patients and greater doses to be tested. 

The cell dosage is highly influenced by the administration route, which usually takes into consideration whether the neurodegenerative pathology to be treated has focal or diffuse lesions within the parenchyma. In focal lesions, the treatment can be an intraparenchymal inoculum, which can be performed with a limited number of cells for reasons of steric hindrance. On the other hand, in multifocal/diffuse diseases, it is possible to use an intrathecal route, typically involving delivery of cells to the subarachnoid space via lumbar puncture or using an intracerebroventricular delivery directly into the cerebrospinal fluid of the cerebral ventricles. The latter two administration routes consent to deliver a higher number of cells maintaining the volume constraint of the liquor spaces.

Not-for-profit phase I clinical trials are particularly challenging, especially with ATMP, since they require coordination of different units in different hospitals and are managed by a small number of people. Our production strategy requires the product to be formulated moments before the administration to the patient; this is quite unusual, since most cell therapy products are infused right after thawing rather than fresh. In our experience, the use of “fresh” suspension of cells is safer for the patients, because it does not imply the administration of any amount of cryopreservative compounds directly in the central nervous system and provides healthy cells, eliminating any affection derived from the freezing procedure. The downside of this approach is the need for a really fine coordination between the operating room and the cell factory as well as the creation of a good control strategy process design and the strict adherence to it. Moreover, a long production process such as ours is quite a big burden for a small cell factory in terms of simultaneous productions. Our process is in fact entirely manual; therefore, each passage requires up to eight hours of work, depending on the size and the phase. Of course, this means that the manual skills of each operator may have a big impact on the process performance; therefore, it could take years to train each person before they become able to operate autonomously. 

In order to move forward to phase II clinical trials both for ALS and MS, due to the higher number of patients to treat, we expanded our production capacity by setting up a bigger cell factory called “Advanced Therapy Production Unit” (UPTA), located in the “Institute for Stem-Cell Biology, Regenerative Medicine and Innovative Therapies” of the “Fondazione IRCCS Casa Sollievo della Sofferenza” research hospital. After a technology transfer of the production process, the UPTA received the authorization for hNSC production for somatic cell therapy (AIFA authorization number 118/2019), and it is also equipped with areas for gene therapy and biopolymer production. Moreover, we are currently working on the scale-up of our process with the implementation of semiautomated systems. This will allow us to perform larger clinical trials in two different hospitals, doubling the production capacity, while at the same time lessening the potential variation in the performance of the process and reducing the time for the trial.

In conclusion, we demonstrated, both in vitro and in vivo, the safety and the ability of hNSCs to be useful in the treatment of neurodegenerative diseases; moreover, we defined GMP-compliant culture conditions based on the use of commercially available serum-free reagents that allow for a better interlaboratory reproducibility of the method.

## 4. Materials and Methods

### 4.1. Tissue Procurement 

The collection of the fetal neural tissues was authorized by the Ethical Committee of the “Fondazione IRCCS Casa Sollievo della Sofferenza” research hospital. Each donor provided written informed consent to the tissue collection after the fetus death. Moreover, a thorough clinical screening to select healthy donors was performed according to the European legislation on tissue and cell donation (Directive 2004/23/EC—Quality and safety standards for donated human tissues and cells). Fetal neural tissue represents the starting material of any production process of hNSCs. As such, each tissue specimen was subjected to quality control evaluation before being used for production.

### 4.2. Production Process 

The production process was conducted as previously described [10,15,16]. Briefly, after collection, the neural tissue was mechanically disaggregated obtaining a single cell suspension. Then, the cells were seeded in a chemically defined medium containing DMEM/F12 (Sigma Aldrich D0547-10L), EGF and bFGF (Peprotech GMP-100-15, GMP-100-18B); this is called complete medium, and it is able to positively select the neural stem cells.

After tissue disaggregation, the production process was conducted in two macrophases:Production of an intermediate product (IP), constituted of cryopreserved human neurospheres from a fragment of fetal neural tissue.Production and delivery of the drug, called final product (FP) in our process, consisting of hNSC suspension, starting from vials of cryopreserved IP.

The drug product (FP) was formulated right before the administration to the final recipient (patient): cells were collected and washed in Hanks’ Balanced Salt solution without phenol red, calcium and magnesium, in order to avoid cell aggregates, and resuspended in the same solution at the final concentration established by the clinical protocol and according to the request of the trial PI. 

### 4.3. Drug Product Characterization

An extensive characterization of the cellular component was established in order to ensure that the cell drug produced was safe and composed of hNSCs according to the release specifications. To reach this goal, we performed the following test.

Identity: identity assessment was conducted both on the starting material and the final product by multiple assays. Regarding the starting material, for each donor tissue, standard hematoxylin/eosin staining was performed on a paraffin-embedded brain specimen to confirm the anatomical origin of the tissue. Another portion of the tissue biopsy was collected and analyzed to verify the gene expression pattern related to positional identity (Table 1). 

Total RNA was isolated using a commercially available kit, following the manufacturer’s protocol (Qiagen, Milan, Italy). As much as 100 ng of the total RNA of each sample was reverse-transcribed. For real-time PCR, each 25 µL reaction mixture consisted of 12.5 µL Universal master mix (Applied Biosystems, Branchburg, NJ, USA) containing dNTPs, MgCl_2_, reaction buffer and AmpliTaq gold, 1.25 µL primers and fluorescent labeled probe specific for each gene (assay-on-demand, Applied Biosystems), and 11.75 µL cDNA diluted in water, corresponding to 100 ng of retro-transcribed RNA. All reactions were performed in duplicate and the mean value was used for calculation of mRNA expression level. Real-time PCR was performed in an SDS 5700 instrument (Applied Biosystems) as follows: 10 min at 95 °C and 40 cycles at 15 s/95 °C and 1 min/60 °C. Data were analyzed with GeneAmp 5700 software (Applied Biosystems). Data obtained were normalized versus GAPDH expression; although we used a commercially available total human brain as internal control, we chose this kind of analysis because of the lack of the specific positive control for each anatomical region of the fetal brain.

For the FP, identity was confirmed by the “in vitro differentiation test” that assesses the ability of the cells to give rise to astrocytes, neurons and oligodendrocytes by immunocytochemical labeling (β-tubulin III, neuron marker; GFAP, astroglial marker; GalC, oligodendroglial marker). Briefly, cells were fixed for 20 min in 4% paraformhaldeyde in PBS, washed, and incubated for 20 min at room temperature with PBS containing 20% normal goat serum (for membrane markers) or PBS/TritonX 0.1% (for intracellular markers), washed again and incubated with primary antibodies overnight at room temperature (GFAP 1:500, bTUB 1:1000, GalC 1:200). Following washing, cultures were incubated for 45 min at room temperature with the secondary antibodies (Alexafluor 1:1000), washed, counterstained with DAPI, and mounted using Fluorsave.

### 4.4. Safety

European Pharmacopeia microbiological tests, genomic stability and tests for adventitious viruses. 

Each cell batch, both IP and FP, was analyzed in order to guarantee microbiological safety and avoid infections or an anaphylactic shock. Tests were performed according to the European Pharmacopoeia requirements for biological sterile products (LAL test, Sterility, Mycoplasma, as described in chapters 2.6.14, 2.6.27, 2.6.21, respectively, of European Pharmacopeia).

Moreover, cells were tested for adventitious virus absence, particularly HSV1/2, both at the beginning and at the end of the production process. Each batch was also tested for genetic stability at four critical control points: starting material collection (brain specimen), I.P. release, second to last passage before DP formulation, DP release. At each time point, we performed karyotype analysis and SNP array.

All these tests were entrusted to qualified contractor laboratories, GMP-qualified when possible.

### 4.5. Biological Tests

Every cell line was tested both in vitro and in vivo to confirm the absence of potential tumorigenic activity. The hNSCs were examined in vitro to verify their growth factor dependence, i.e., their inability to replicate themselves in the absence of EGF and bFGF as previously described [10]. In the in vivo testing, the nontumorigenicity was verified using immunodeficient nude mice (Athymic Nude-Foxn1nu): briefly, hNSCs (300.000 cells/animal, *n* = 7 animals/cell line) were injected into the brain (striatum, AP:0, L: −2.5, DV: −2.7) and sacrificed 6 months after transplant; histological analysis on cryopreserved brain coronal sections was performed to verify the presence of abnormal proliferation and the ability of the cells to integrate into the host tissue (See ref [10] for protocol details). The bio-distribution of the human stem cells was evaluated through the use of the anti-human nuclei antibody (huN, Chemicon, MAB1281 1:200), followed by a secondary antibody (anti Mouse IgG, Alexa Fluor 488, Jackson, 115545205, 1:1000), in the whole brain and peripheral organs (heart, lungs, spleen, kidneys, intestine) at 6 months after transplant. The proliferating fraction of the hNSCs cells was determined by use of antibodies recognizing Ki67 (Novus Biological NB600-1252, 1:500) and Alexa Fluor 594 (Invitrogen, A32740 1:1000). The quantification of proliferating cells was determined by counting huN^+^/Ki67^+^ cells over total huN^+^ cells in serial coronal sections of the mouse brain containing the graft.

### 4.6. Purity

The DP formulation requires multiple washing steps in order to remove as many process-related impurities as possible (i.e., medium component); to validate this process, during the DP formulation for each washing step made with the HBSS, the supernates were analyzed using a reverse-phase HPLC analyzer and compared with both the complete medium and the HBSS itself. 

The analytical HPLC was performed on each sample, with a Waters 1525 Binary pump fitted with a Waters 2487 dual λ Absorbance UV/Vis detector, using a Restek™ analytical C18 column (250 × 4.6 mm, 5 μm); flow rate 1 mL min^−1^; λ = 240 nm and λ = 257 nm, with a gradient elution of acetonitrile (AcN) in distilled water (H_2_O) with 0.1% trifluoroacetic acid (TFA) over 50 min (A = 0.1% TFA in H_2_O, B = 0.1% TFA in AcN). All samples were used without further solvent treatments (20 μL total volume). Each experiment was conducted in quadruplicate from three different lots. Data processing was performed with the Breeze™ and the Origin 8™ software. Product-related impurities, such as debris or nonvital cells, were evaluated through the culture and at the final formulation using the trypan blue exclusion test as prescribed by the European Pharmacopoeia.

### 4.7. Potency

#### 4.7.1. In Vitro 

In vitro potency was assessed by evaluating two stemness-related functional properties of each cell-line: (i) the self-renewal potential, and (ii) the clonogenic ability. The first characteristic was evaluated by calculating the growth kinetics of the cells during long-term culture (not less than 5 amplification passages), while the clonogenic capacity of the cell population was estimated using the “clonal efficiency assay”. Both protocols have been previously described [10]. 

#### 4.7.2. In Vivo 

Potency was also evaluated in vivo, assessing the therapeutic potential of the hNSCs in the SOD1 rat model of ALS. 

In order to evaluate the symptomatic hallmarks of ALS, together with the markers of neuroinflammation, the hNSCs were transplanted in SOD1 rats using the same microsurgical approach used for the ALS patients. See Zalfa et al. for full protocol description and results [17].

## Figures and Tables

**Figure 1 ijms-23-13425-f001:**
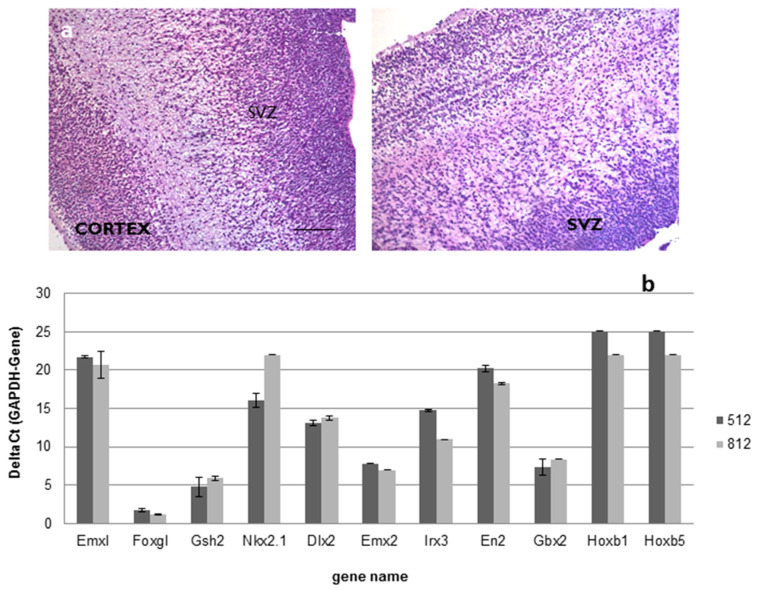
(**a**,**b**). Positional identity: ematoxylin and eosin staining followed by histological examination confirmed the subventricular origin of brain sample used as raw material in the hNSCs production (Scale bar: 100 µm) (**a**); real-time PCR analysis confirmed that all the samples used in the production belonged to the forebrain (**b**).

**Figure 2 ijms-23-13425-f002:**
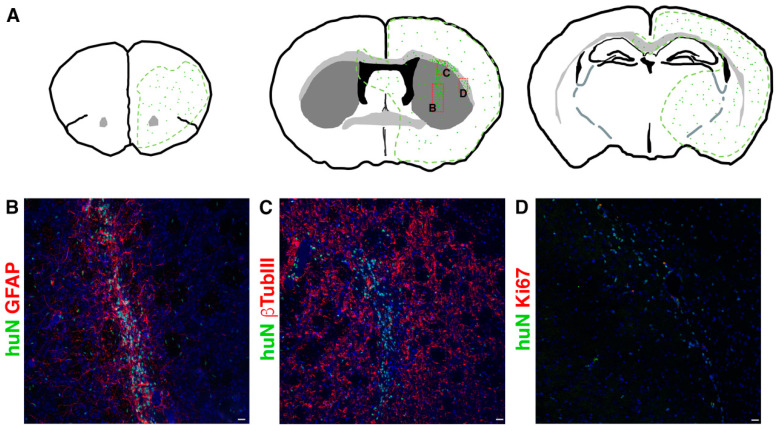
(**A**) Schematic showing typical migration patterns of hNSCs transplanted into the striatal region of the brain ((**A**), middle). The area outlined in green represents the main brain areas reached by hNSCs (symbolized by green dots). Red boxes indicate the position of pictures showed in (**B**,**C**,**D**) respectively. (**B**–**D**). Transplanted hNSCs, huN^+^ (green in (**B**–**D**)), engraft the striatum (graft core in (**B**,**C**)) and migrate in the surrounding parenchyma (single huN^+^ cells in (**B**,**C**)) and along the corpus callosum (**D**). The cells express astrocyte markers (GFAP, red in (**B**)), neuronal markers (bTubbIII, red in (**C**)) and the proliferation marker Ki67 (red in (**D**)). Scale bar in (**B**–**D**): 20 µm.

**Figure 3 ijms-23-13425-f003:**
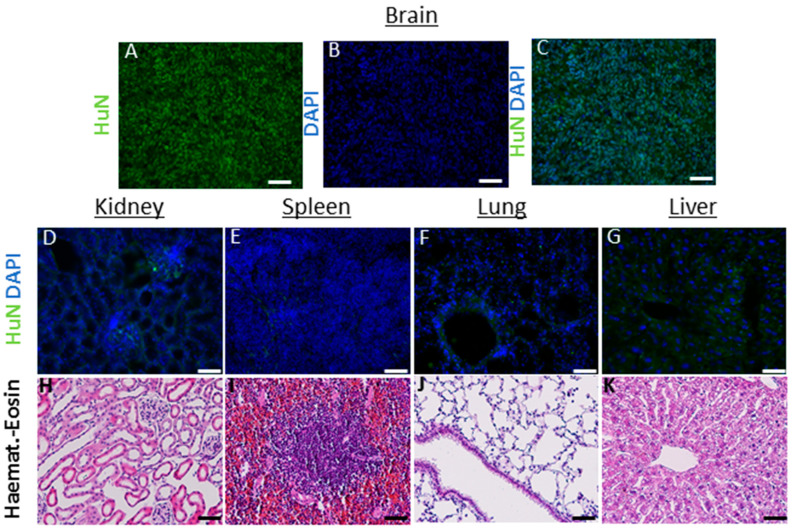
(**A**–**K**) Analysis in nontarget organs: the antibody huN, which selectively recognizes only the nucleus of human cells, was used to detect the presence of transplanted hNSCs, within the parenchyma of mouse organs outside the brain (6 months after transplant). Brain sections containing human cancer neural stem cells (**A**–**C**) were used as positive controls to assess the correct functioning of the antibody and immunofluorescence protocol. No cells were detected in kidney (**D**), spleen (**E**) lungs (**F**) and liver (**G**) and lymph nodes (not shown). As additional safety evaluation, hematoxylin eosin was used to assess the preservation of the parenchyma, the absence of inflammatory processes (edema, blood-born immune cells infiltration) or signs of de novo tumorigenesis (**H**–**K**). Scale bars: (**D**–**G**) 50 µm. (**H**–**K**) 100 µM.

**Figure 4 ijms-23-13425-f004:**
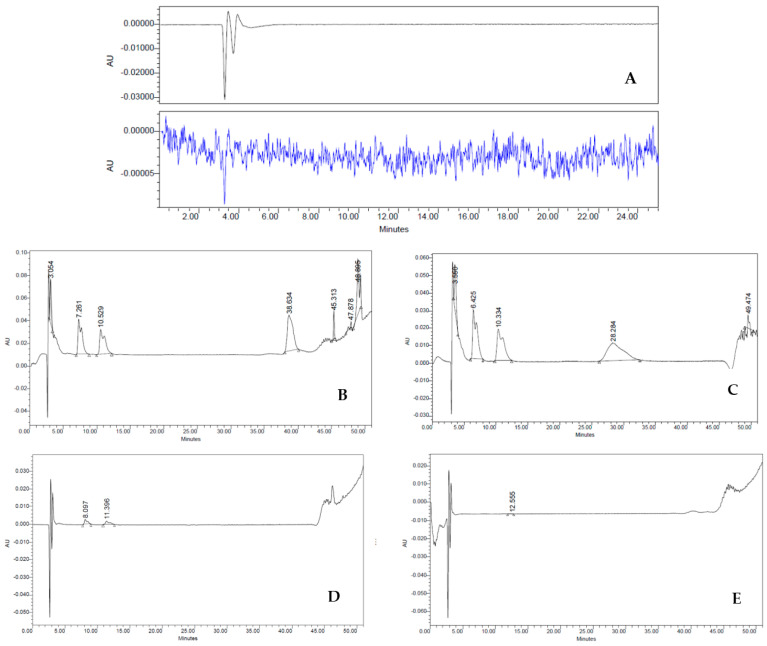
(**A**–**E**) HPLC analysis: Three major fractions were observed with an RT of 6.47 min, 10.36 min and 28.28 min (**A**). HBSS characteristic baseline over the 50 min (**B**). Worn-out medium decreasing amounts of hydrophilic fraction (**D**). First wash (**E**). Second wash sample drastic change in the shape of the HPLC spectrum (**E**) that became similar to the one observed in the HBSS.

**Table 1 ijms-23-13425-t001:** Positional identity: gene and primers used to identify the brain region of the tissue samples.

Gene	Region of Expression	Primer
Foxg1	Prosencephalon	F:GGGCAACAACCACTCCTTCTCCAC R:GACCCCTGATTTTGATGTGTGAAA
Emx1/2	Cortex/olfactory bulb	F:GCGGCCTTCGTGAGTGGCTT R:GGATCCGCTTGGGCTTGCGT
Gbx2	Hindbrain	F:CCTGGCCAAAGAGGGCTCGC R:GCTGCTCGCTGGTGAAGGCA
Gsh2	Telencephalon	F:GCAGCACCACGCACCTGTCT R:CCTGGCTCCCGACGCACTTG
Irx3	Diencephalon	F:CGGAGACTGCCACAAGCCCG R:ATGCGGGGTGCCACAGAAGC
Hoxb5	Spinal cord	F:CCTCCAGCCACTTTGGGGCG R:CGGACAGGCAGAGTGCGTGG
Hoxb1	Spinal cord—R4	F:GCGCCCCAACCTCCTTTCCC R:GGTGGCGGCAATCTCCACCC
En2	Cerebellum	F:TCGGACTCGGACAGCTCGCA R:GGCCCCTCCTGCTGTCCTCA
Dlx2	Subpallium	F:GCGTACACCTCCTACGCGCC R:GAAGCGCTGGCTCCAGGGTG
Nkx2.1	Forebrain	F:AAAGTGGGCATGGAGGGCGG R:CCGCCTTGTCCTTGGCCTGG

## Data Availability

Not applicable.

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
