# Peer review of "Human Neural Stem Cell-Based Drug Product: Clinical and Nonclinical Characterization"

_ijms, 2022, doi:10.3390/ijms232113425_

Round 1
Reviewer 1 Report
The manuscript presented from Profico et al., is highly interesting, original and well structured. My concern is related to the qPCR, why the authors used delta CT and not delta delta ct?
again is not clear if there are significant difference, the authors should show more detail concerning the raw data.
Author Response
Referee #1 (Remarks to the Author):
The manuscript presented from Profico et al., is highly interesting, original and well structured. My concern is related to the qPCR, why the authors used delta CT and not delta delta ct?
again is not clear if there are significant difference, the authors should show more detail concerning the raw data.
AUTHORS' RESPONSE:
We thank the reviewer for the positive appraisal of our manuscript.
We add explanation on CT use on page 10 line 399.
Essentially we choose this kind of analysis because there are not positive controls for each single region of the fetal brain and we are forced to use a commercially available total human brain extract. We didn’t show a statistical analysis because here we are just presenting the data about the two tissues we used for the clinical trials so it would be unconclusive to perform any analysis. Nevertheless we presented raw data from the real time PCR that you can find in the supplementary materials, table S.1

Reviewer 2 Report
The authors describe the quality control strategy for the characterization of the hNSCs used in the above-mentioned clinical trials. While I admit their effort and its importance, there are some issues to be resolved.
1. They should show the summary table of test results on each sample, not just examples.
2. Although "Data collected from all the productions performed in our facility show that the population obtained with our differentiation protocol is composed by 44,57±11,99% astrocytes, 24,91±11.14% neurons and 20,75±7,98% oligodendrocytes.", there are no detailed description about the methods and data of their characterization.
3. The data and characters in Figure 3 are too small to read.
4. The words "DP batches" and "IP batches" are not defined in this manuscript. They should define it first.
Author Response
Referee #2 (Remarks to the Author):
The authors describe the quality control strategy for the characterization of the hNSCs used in the above-mentioned clinical trials. While I admit their effort and its importance, there are some issues to be resolved.
- They should show the summary table of test results on each sample, not just examples.
- Although "Data collected from all the productions performed in our facility show that the population obtained with our differentiation protocol is composed by 44,57±11,99% astrocytes, 24,91±11.14% neurons and 20,75±7,98% oligodendrocytes.", there are no detailed description about the methods and data of their characterization.
- The data and characters in Figure 3 are too small to read.
- The words "DP batches" and "IP batches" are not defined in this manuscript. They should define it first.
AUTHORS' RESPONSE:
We are grateful to the reviewer for the careful and fair appraisal of our manuscript.
1 We add complete results for differentiation test, in vivo analysis and positional identity in the supplementary material, table S.1, S.2 and S.3
2 We add detailed descriptions on page 10 about the method and complete results in supplementary materials.
3 We ameliorate the quality and definition of figure 3 (now figure 4)
4 We revised all acronyms and made them explicit.

Reviewer 3 Report
In this communication manuscript, the authors describe the procedure of hNSPCs characterization for preclinical and clinical trials. I have the following concerns:
Evidence regarding the fates of transplanted hNSPCs in vivo are very limited in this work.
It seems that there are some cells stained with HuN in Figure 2, more data are needed exclude the existence of transplanted cells in other organs.
“Neural stem cells (NSCs)” to ‘neural stem/progenitor cells (NSPCs)’
Line 33: Please provide full names for GMP and ATMPs;
Please reorganize the Introduction for a better logic.
Line 140: Please check the meaning of ‘average 35,6±9,46’.
Line 153: Please correct the typo ‘hNCSs’.
Line 181: Please check the sizes of scale bars.
Please discuss the limitations and challenges of fetuses-derived hNSPCs in the treatment of neurological diseases.
Author Response
Referee #3 (Remarks to the Author):
In this communication manuscript, the authors describe the procedure of hNSPCs characterization for preclinical and clinical trials. I have the following concerns:
Evidence regarding the fates of transplanted hNSPCs in vivo are very limited in this work.
It seems that there are some cells stained with HuN in Figure 2, more data are needed exclude the existence of transplanted cells in other organs.
“Neural stem cells (NSCs)” to ‘neural stem/progenitor cells (NSPCs)’
Line 33: Please provide full names for GMP and ATMPs;
Please reorganize the Introduction for a better logic.
Line 140: Please check the meaning of ‘average 35,6±9,46’.
Line 153: Please correct the typo ‘hNCSs’.
Line 181: Please check the sizes of scale bars.
Please discuss the limitations and challenges of fetuses-derived hNSPCs in the treatment of neurological diseases.
AUTHORS' RESPONSE 1:
We are grateful to the reviewer for the careful and fair appraisal of our manuscript.
We add figure 2 to better explain the distribution of hNSCs in brain of nude mice and we ameliorate the resolution of figure 3 (before fig.2) panel A (single channel here attached) to highlight that the green color that is seen is mainly due to the autofluorescence of the tissue and its intrinsic ability to retain dye that generates non-specific signal.
We revised all acronyms and made them explicit, correct the typo in line 153 and eliminate average 35,6±9,46 in line 140 that was another typo.
We reorganised the introduction, moreover we discuss the greater difficulties in using hNSCs in neurological disease treatment in discussion session, page 8 line 291.
We have used the designation hNSCs in all previously published manuscripts and the cellular drug itself that we describe in this work has this designation also in all documents released by regulatory authorities. For that reasons we would like to keep hNSCs denomination.
We revised the figure caption and corrected the scale bar which was altered due to a typo.

Round 2
Reviewer 1 Report
The authors satisfied all my concerns
Reviewer 2 Report
The authors revised the manuscript sufficiently.